# Milk and resistance exercise intervention to improve muscle function in community-dwelling older adults at risk of sarcopenia (MIlkMAN): protocol for a pilot study

Antoneta Granic ,[1,2,3] Christopher Hurst,[1,3] Lorelle Dismore,[1,3] Karen Davies,[1,2,3] Emma Stevenson,[4,5] Avan A Sayer,[1,2,3] Terry Aspray[3,4]

CH and LD contributed equally.

For numbered affiliations see end of article.

**Correspondence to**
Dr Terry Aspray;
terry.aspray@newcastle.ac.uk

## ABSTRACT

**Introduction** Sarcopenia is a progressive muscle disorder characterised by decline in skeletal muscle mass, strength and function leading to adverse health outcomes, including falls, frailty, poor quality of life and death. It occurs more commonly in older people and can be accelerated by poor diet and low physical activity. Intervention studies incorporating higher dietary protein intakes or protein supplementation combined with resistance exercise (RE) have been shown to limit muscle function decline. However, less is known about the role of whole foods in reducing the risk of sarcopenia. Milk is a source of high-quality nutrients, which may be beneficial for skeletal muscle. This pilot study examines the feasibility and acceptability of milk consumption with RE to improve muscle function in community-dwelling older adults at risk of sarcopenia.

**Methods and analysis** 30 older adults aged ≥65 years will be randomly allocated to three groups: 'whole milk+RE', 'skimmed milk+RE' or 'control drink+RE'. Assessments will take place in participants' homes, including screening (milk allergies, grip strength, walking speed), baseline and postintervention health and function. All participants will undertake a structured RE intervention twice a week for 6 weeks at a local gym, followed by the consumption of 500 mL of whole or skimmed milk (each ~20 g of protein) or an isocaloric control drink and another 500 mL at home. Participants' views about the study will be assessed using standardised open-ended questions. The primary outcomes include feasibility and acceptability of the intervention with recruitment, retention and intervention response rates. Analyses will include descriptive statistics, exploration of qualitative themes and intervention fidelity.

**Ethics and dissemination** Outputs include pilot data to support funding applications; public involvement events; presentation at conferences and peer-reviewed publication.

**Trial registration number** ISRCTN13398279; Pre-results.

## Strengths and limitations of this study

► To our knowledge, this is the first pilot study examining the feasibility and acceptability of whole versus skimmed milk with resistance exercise (RE) intervention in community-dwelling older adults living in the UK.

► The intervention is conducted in a local gym that is easily accessible to older adults who will benefit from the familiarisation with RE programme conducted in the community to foster continuous engagement.

► Postintervention interview will allow for the collection of qualitative evidence to support planned future trial, including better understanding of the barriers and facilitators of community-based intervention.

► Because this an evaluation of a pilot implementation, the sample size is not based on statistical power.

► Although we do not anticipate any definite results in exploring differences between intervention groups, the results will be used to aid power calculations for planned future substantive research.

## INTRODUCTION

The UK population is ageing rapidly; the number of adults aged ≥65 increased by 17.3% in the last decade, and in mid-2017 was estimated to account for 18.2% of the total population of 66 million[1]. Understanding factors associated with healthy ageing[2] such as diet and physical activity[3] for optimising health and well-being of an ageing population is essential for the development of effective interventions.

Sarcopenia is a progressive, generalised muscle disorder characterised by decline in skeletal muscle mass, strength and function,[4] which leads to an increased risk of falls, frailty, disability, low health-related quality of life (QoL) and death.[5–8] It occurs more commonly in older people and can be accelerated by poor diet and low physical activity.[8 9] The prevalence of sarcopenia increases with advancing age—and although dependent on the algorithm used to define sarcopenia[10]—it

reaches >20% in men and women aged ≥85 years,[11] resulting in estimated healthcare costs in excess of £2.5 billion/year in the UK.[12] This emphasises the need for sustainable preventive measures aimed to preserve and optimise muscle health and function in a rapidly ageing population before the onset of difficulties leading to or exacerbating the risk of sarcopenia.

### Protein intake and exercise for healthy muscle ageing

Loss of muscle mass and strength can be accelerated by poor diet, low levels of physical activity and the presence of long-term conditions,[8 9 13] leading to diminished QoL.[7 14] Adequate intake of dietary protein and resistance exercise (RE) are recognised as effective interventions to promote skeletal muscle health and reduction of physical decline.[15 16] Specifically, intervention studies that examined a combined effect of protein supplements, comprising essential amino acids (EAA) and RE to stimulate muscle protein synthesis (MPS) have observed an increase in total muscle protein within 3–5 hours following exercise in both young and older adults.[17] Compared with young adults, older adults experience a blunted response after protein ingestion to stimulate MPS (anabolic resistance), especially in response to lower amounts of protein or EAA of <20 or <10 g, respectively. Other studies have shown that greater amounts of protein supplementation and intermittent feeding in combination with repeated bouts of RE resulted in increased muscle mass in older adults, even in those diagnosed with frailty and sarcopenia.[18–20]

However, there is limited research on the role of whole foods rich in protein (eg, milk and dairy products, fish and meats) in maintaining skeletal muscle mass, strength and function in older adults at risk of sarcopenia. Regular consumption of high-quality, nutrient-dense foods, high in macronutrients and micronutrients relevant for muscle[21] within a varied diet may provide a platform for developing strategies for maintenance of muscle health and function in later life that do not include supplements and medical products, and may be easier adopted as a behavioural change in older adults.[22]

### Milk for muscle health: current evidence and why this pilot is needed

Cow's milk is an example of a whole food with the potential to ameliorate loss of skeletal muscle mass, strength and function. Whole milk (3.6% fat) is a source of high-quality proteins (whey and caseins), minerals (eg, calcium, phosphorus, magnesium), vitamins (eg, A, B, D and E), carbohydrates, bioactive lipids and fatty acids (monosaturated and polyunsaturated, and saturated fatty acids).[23] Whey protein is considered superior to other protein sources for MPS after exercise in younger and older adults because of its greater bioavailability and solubility, and higher content of the branched-chain amino acids, including leucine.[24–26] Furthermore, the concurrent intake of milk fats with protein in whole milk has been shown to increase the use of EAA for MPS after

exercise in young men compared with skimmed milk (0.3% fat),[27] suggesting additional benefits of milk lipids for muscle. Other benefits of milk containing fat include reduction in exercise-related muscle damage, soreness and decline in muscle performance in young adults and athletes[28 29] compared with energy-matched (isocaloric) carbohydrate drink. However, little is known about the effect of milk and protein-fat ratio in milk on muscle in older adults, particularly the impact on muscle function of varying milk fat contents (whole vs skimmed) providing >20 g protein/day after exercise.

We hypothesised that whole milk (3.6% fat), providing >20 g of protein and the same amount of energy as fat and protein-free carbohydrate drink, after structured exercise conducted in the community may be a feasible and acceptable intervention for maintaining skeletal muscle mass, strength and function in older adults at risk of sarcopenia.

### Study aims

The primary aims are:

1. To examine the feasibility and acceptability of whole (3.6% fat) or skimmed milk (0.3% fat) in combination with RE as an intervention in community-dwelling older adults aged ≥65 at risk of sarcopenia. This aim will answer the following questions: Is an intervention of 2×500 mL milk+RE twice a week for 6 weeks (a) feasible and (b) acceptable to older adults?
2. To provide essential data for planned future substantive research.

The secondary aim of the study will be to explore whether consumption of whole or skimmed milk+RE has an influence on physical performance, muscle mass, strength and self-reported QoL in older adults at risk of sarcopenia.

## METHODS AND ANALYSIS

### Study design

This is a pilot study with a parallel-group design involving 30 participants (aiming for 15 men and 15 women) aged ≥65 years who will be randomised into three intervention groups: (group 1) 'whole milk+RE'; (group 2) 'skimmed milk+RE' and (group 3) 'control drink+RE'. Data will be collected from: (1) health and functioning assessments (screening, baseline and postintervention interview); (2) the nutrition+exercise intervention over 6 weeks and (3) participants' feedback about the study.

### Exclusion and inclusion criteria

The study will include older adults who are registered patients with general practitioner (GP) practices within the National Institute for Health Research (NIHR) North East and North Cumbria Clinical Research Network (CRN), UK. Table 1 lists the inclusion and exclusion criteria that are applied to patient database searches, performed in GP practices and screening interviews conducted by the research team.

**Table 1** Inclusion and exclusion criteria for the MIlkMAN: pilot

| Criteria | Patient database searches | Screening interview |
|---|---|---|
| Inclusion | | |
| | Aged 65 years and over | |
| | Live in the community | |
| Exclusion | | |
| | Diabetes mellitus type 1 or type 2 | Lacks capacity to consent to participate |
| | Chronic kidney disease stage 4 or 5 (estimated glomerular filtration rate<30 mL/min/1.73 m$^2$) | Lactose intolerance |
| | Liver function impairment (AST>2.5 times upper limit of normal range within the last 6 months) | Dislikes milk or cranberry juice (control drink) |
| | | Participated in a structured RE training and gym programme in the last month |
| | Chronic lung disease requiring maintenance steroid therapy (eg, COPD, severe asthma) | Dislikes gym exercise with equipment |
| | End-stage terminal illness | Unintentional weight loss ≥5 kg in the last 3 months |
| | Cardiac pacemaker or severe heart failure or other significant heart disease | Unable to understand instructions for muscle strength and function assessments in English or unwilling to participate in protocol when explained |
| | Uncontrolled hypertension (>160/100 mm Hg) and uncontrolled hypotension (<100 mm Hg systolic) within last 6 months | An individual who the research team (exercise physiologist) evaluates as not suitable for the intervention because of safety reasons |
| | Hip or knee replacement | |
| | Impaired mobility (unable to walk without an aid including wheelchair) | |
| | Current prescription of warfarin (potential interference with control drink) | |
| | BMI≥30 kg/m$^2$ | |
| | An individual who the GP feels it is inappropriate for the research team to approach for safety reasons: any medical and physical conditions that preclude safe participation in a RE programme (long-term conditions likely to lead impaired function over 6 months) | |

AST, aspartate aminotransferase; BMI, body mass index; COPD, chronic obstructive pulmonary disease; GP, general practitioner.

## Study population and recruitment

Primary care recruitment will be carried out with the assistance of North East and North Cumbria CRN, England, which provides support with access to general practices and their patients. Two practices in the North Tyneside Clinical Commissioning Group were identified for feasibility using exclusion/inclusion criteria (table 1, left column) and provided feedback during the application for funding in 2018.

Recruitment will be organised in two stages: prescreening (GP practices) and screening (research team). At the prescreening stage, practices will identify potential participants from their patient database using exclusion/inclusion criteria (table 1, left column), and then mail out recruitment packs, containing detailed information about the study with a reply slip. Interested individuals will be interviewed over the telephone by a researcher using a 5-item SARC-F questionnaire (online supplementary appendix 1)[30] to assess any difficulties with day-to-day activities (lifting and carrying 10 pounds,

walking, rising from a chair and climbing stairs), and number of falls in the past year. Those evaluated by the research team to have no major difficulties that would preclude safe participation in the exercise programme (eg, unable to walk across a room), will be visited in their own home for a screening interview to obtain written informed consent, and to evaluate inclusion/exclusion criteria not screened through GPs (table 1, right column).

Those who meet the criteria will be assessed further for muscle strength (grip strength (GS) and function (walking speed) based on the following cut-offs: <20 kg (women) and <30 kg (men)[31] for low GS; and <0.8 m/s or ≥5 s over 4 m distance[31] for low walking speed. GS measurements (high or low) at the screening interview will be used for minimisation along with sex to allow equal distribution of those with muscle strength weakness across the intervention groups. However, the target number of those with 'low' GS will not be established a priori. Therefore, the study will recruit older adults with some deficits in muscle health and those without for whom it is determined to

**Table 2** Study outcome measures

| Measure | Screening | Baseline | Postintervention |
|---|:---:|:---:|:---:|
| **Primary** | | | |
| Feasibility and acceptability of intervention in a local gym setting | | | × |
| Applicability | | | × |
| Dosage and duration of intervention | | | × |
| Compliance | | × | × |
| Attrition | | × | × |
| Adverse health effects | | | × |
| Response rates to questionnaires, assessments and intervention | × | × | × |
| **Secondary** | | | |
| Short physical performance battery[63] (balance, 4 m gait speed, 5-chair stands) | | × | × |
| Muscle mass[33] | | × | × |
| Grip strengh[32] | × | × | × |
| SF-12 Health Survey[34] | | × | × |
| Barthel ndex[35] | | × | × |

SF-12, Short Form 12-item Health Survey.

be safe to participate in the study (primary aims), and hypothesised to benefit from the intervention regardless of the deficits (secondary aims).

### Study outcomes

Table 2 lists the primary and secondary study outcomes and when they are completed.

The secondary outcome measures will explore differences in physical performance measures between the groups, preintervention and postintervention. GS (muscle strength)[32] will be expressed as the maximum reading of six trials of both hands using a Jamar hand-held 5030J1 dynamometer. Body composition (muscle mass) will be measured using Bioelectric Impedance Analysis[33] (BIA; Tanita MC-780MA Body Composition Analyzer). Self-reported QoL will be measured using 12-item Short Form Health Survey (SF-12),[34] and activities of daily living with Barthel Index.[35]

### Randomisation

A researcher will allocate 30 participants to one of the three interventions to ensure balanced allocation of participants between the groups based on sex and muscle strength (GS at screening assessment) using a free, open-source minimisation software (MiniPy 0.3, http://minimpy.sourceforge.net).[36] The software features elements of randomness in the minimisation algorithm by allocating the first participant randomly into one of the interventions, and assigning the subsequent participants on hypothetical stepwise allocation to every group and computation of the imbalance score corresponding to each allocation. The imbalance scores are compared and participants allocated to the group corresponding to the least imbalance score (preferred group).

The sample size for the pilot is not based on statistical power but guided by the consideration to fulfil the primary aims of the study (eg, provide guidelines for the larger trial) and practical feasibility.[37]

### Consent

Written informed consent will be obtained by a researcher visiting the participants during the home-based screening assessment prior to randomisation. Capacity to consent will be assessed using an established consent pathway. Throughout the active research phase (ie, from baseline to postintervention assessment), the notion of process consent will be implemented, requiring an ongoing exchange of information about the study and confirming the participants willingness to proceed, ensuring that participants are free to reconsider and withdraw from the study at any time. If a participant loses capacity during the research process, he/she will be withdrawn from the study.

### Flow diagram of the study

A flow diagram of the study protocol with timelines is outlined in figure 1. The recruitment (assessment of eligibility) up to allocation (randomisation into three groups) will be finalised within 3 months, followed by baseline assessments for health and functioning in participants' homes, and a 6-week intervention in a local gym.

Home-based postintervention assessments, including participants' feedback about the study, will be conducted the week following completion of the intervention, and finalised within 3 weeks. Data analysis will be completed after active data collection (from randomisation to postintervention assessment). Data collection for each participant will span approximately 10 weeks: (1) week 1: consent and home-based screening assessment; (2)

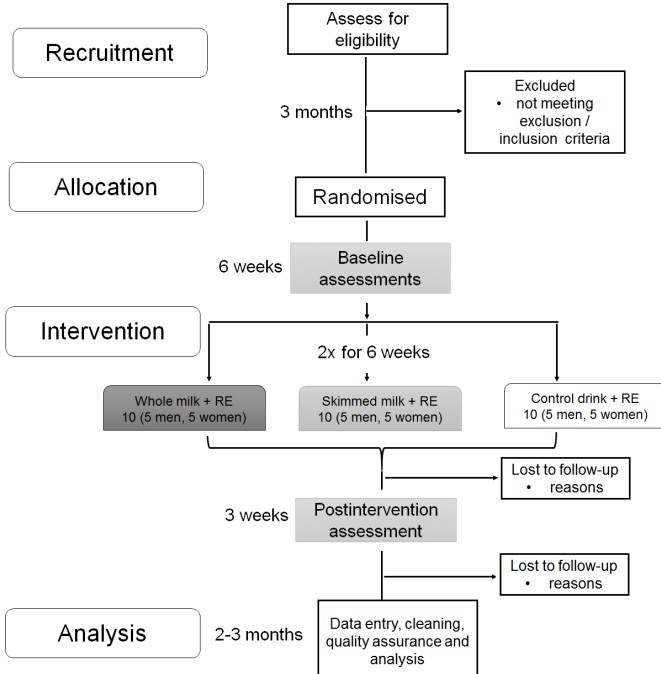

**Figure 1** Study flow chart. The recruitment (assessment of eligibility) up to allocation (randomisation into three groups) will be finalised within 3 months, followed by baseline assessments for health and functioning in participants' homes, and a 6-week intervention in a local gym. A postintervention assessment will be conducted over 3 weeks in participants' homes. Data will be analysed following data entry, cleaning and quality assurance over 2–3 months.

week 2: home-based baseline interview; (3) weeks 3–8: 6-week intervention twice a week (12 visits at a local gym/ sport centre); (4) week 10: postintervention home-based interview. Except for the intervention (six consecutive weeks), this time scale can be adjusted to participants' individual needs with a maximum 3 weeks gap between baseline assessment and the first week of intervention, and a maximum 3 weeks gap between the last week of intervention and postintervention assessment.

### Data collection

To determine the feasibility and acceptability of the study, the following data will be collected and analysed: the number of individuals approached; the reasons for not opting to take a part in the study (reported with permission); the recruitment and retention rates; the reasons for attrition; the completion of objective assessments and questionnaires; the number of RE sessions completed and compliance with the milk/control drink intake. Other health and functioning data will be collected at the home-based screening and baseline interview, during the intervention (at the gym), and at the home-based postintervention interview. Participants' attitudes and opinions about the study will be collected at the postintervention interview using a combination of multiple-response and standardised open-ended questions.

### Screening interview

Once potential participants have been identified by the GP practices, having expressed an interest in participation and being interviewed over the phone by a researcher, a mutually convenient appointment will be arranged for a screening visit at the participant's home. Participants will be screening for other exclusion/inclusion criteria not assessed at by the GP practices (table 1, right column), and to establish participants' muscle strength (GS) and functioning status (walking speed) based on the established cut-offs.[31] Informed consent will be obtained before any assessment is undertaken. Eligible individuals will be informed about the study procedure and their journey through the study (from randomisation to postintervention assessment).

### Baseline interview

Table 3 lists domains and assessments for the baseline interview with times needed to administer (in minutes). The detailed health and functioning profile will involve minimal risk and inconvenience to participants, and it will be conducted within 70 min (including breaks and excluding assessments performed at the gym).

### Intervention
#### Resistance exercise

All participants will perform two RE sessions per week for 6 weeks at a community leisure centre (The Parks, North Tyneside Council, North Shields, UK) that is easily accessible and close to their residence. For each RE session a time slot of ~45–60 min in duration will be allocated, with a minimum of 48 hours between sessions. The sessions will be completed in groups of two to four participants under the supervision of an experienced exercise physiologist (CH). Exercise intensity, volume, frequency and duration have been determined based on recent literature[38–40] and the American College of Sports Medicine recommendations for older adults.[39] With the exception of the structured exercise sessions and the nutritional intervention prescribed, participants will be asked to maintain their usual diet, level of physical activity and lifestyle throughout the duration of the intervention period.

During the first RE session, participants will be familiarised with the exercises (leg press, leg curl, seated row, chest press) as well as the equipment to be used throughout the intervention with correct technique demonstrated and extensively described. Following this, participants' one repetition maximum (1RM) will be estimated for all four exercises using a previously established equation.[41]

Following the initial RE session, each remaining session will begin with a 5 min warm-up performed at progressive intensity using either a cycle ergometer or treadmill. Participants will then complete 2–4 sets of 8–12 repetitions at a workload of 70%–79% 1RM[38 39] for all four of the exercises listed above. Each session will conclude with a short cooldown period of low-ntensity aerobic exercise, and (except the initial session) may be completed within 30 min.

**Table 3** Domains and assessments at baseline

| Domain and assessment | Time to administer (min) |
|---|---|
| Sociodemographic profile | Total: 6 |
| Age | |
| Sex | |
| Marital status | |
| Education | |
| Social class (NS-SEC)[64] | |
| Deprivation (Multiple Index of Deprivation)[65] | |
| General health | Total: 54 |
| SF-12 Health Survey[34] | 4 |
| Self-reported diseases diagnosed by a doctor | 2 |
| List of medication (prescribed and over-the-counter) | 2 |
| Mini Mental State Examination | 10 |
| Geriatric Depression Scale (15-item version)[66] | 7 |
| Barthel Index (activities of daily living)[35] | 3 |
| Blood pressure (systolic and diastolic)* | 5 |
| Intake24: 24 hours dietary recall[67]* (https://intake24.co.uk/) | 20 |
| Appetite (a 4-item Simplified Nutritional Appetite Questionnaire)[68]* | 1 |
| Lifestyle | Total: 5 |
| Self-reported physical activity[69] | 3 |
| Smoking status | 1 |
| Alcohol intake | 1 |
| Anthropometry | Total: 14 |
| Demi-span | 2 |
| Waist and hip circumference | 3 |
| Calf circumference | 2 |
| Muscle mass (body composition by BIA)[33]* | 7 |
| Physical functioning | Total: 24 |
| Short Physical Performance Battery[63] | 10 |
| Balance (a side-by-side tandem; semi-tandem; tandem) | 3 |
| 4 m gait speed | 3 |
| 5-chair stands | 4 |
| Maximum grip strength (measured three times in each arm) | 4 |

NS-SEC, The National Statistics Socio-economic classification (Office for National Statistics, UK).
*Assessments done at the intervention site (gym) before and after each RE session (blood pressure), and before the first RE session (diet, appetite, body composition).
BIA, Bioelectric Impedance Analysis; RE, resistance exercise; SF-12, 12-item Short Form Health Survey.

In an attempt to promote participants' engagement with RE, each will receive a training log with diagrams and short instructions with space to record the details of the exercise completed. Participants' gym attendance, sets and repetitions completed and weight lifted will be recorded following each RE session allowing for the calculation of measures of training load (eg, volume load (number of sets×number of repetitions×weight lifted)). In addition to measures of external training load, resistance training intensity will be monitored using participant ratings of perceived exertion. Using the CR100 scale[42] (online supplementary appendix 2), participants will provide an overall session rating of perceived exertion (sRPE) as well as differential ratings of perceived exertion for upper-body muscle exertion (RPE-U) and lower-body muscle exertion (RPE-L) approximately 10 min after the completion of each RE session.[43] Each participant must complete at least 10 sessions (out of 12) to be considered compliant with the exercise programme.

Blood pressure and heart rate will be measured pre and post each RE session in each participant and compared with the guidelines provided by the American College of Cardiology/American Heart Association Task Force[44] and existing literature.[45 46] Muscle soreness will be assessed using a simple visual analogue scale (online supplementary appendix 3) at ~40–45 min and at ~6–7 hours after each RE session.

### Nutritional intervention

On average, 500 mL milk contains ~20 g of protein needed to stimulate MPS above stimulation provided by RE.[17 18] Whole cow milk (nutritional estimates of 22 UK samples during winter and summer) provides 66 kcal/100 g of energy.[47] Arla Cravendale whole milk contains 3.6 g fat, 3.4 g protein and 4.7 g of carbohydrates per 100 g of milk. Arla Cravendale skimmed milk contains 0.3 g fat, 3.6 g protein and 4.9 g of carbohydrate per 100 g of milk. The energy content of the control drink (cranberry juice; Ocean Spray Classic 23/kcal/100 g of energy) will be balanced to match whole milk energy content and supplemented with maltodextrin (4 kcal/g; www.myprotein.com) on the day of intervention. Both milk and juice will be provided in packs of 1 L, bought fresh on a weekly basis through a local retailer and kept in a locked refrigerator at the Campus for Ageing and Vitality, Newcastle University.

The milk/control drink will be consumed as a bolus intake of 500 mL under the supervision of a researcher immediately after exercise during the recovery period, aiming for complete consumption within ~45 min prior to leaving the centre. The second dose of 500 mL will be consumed at the participants' home over the next 4–5 hours as a part of their usual diet with other foods. Participants' compliance with consumption of the milk/control drink will be checked in the evening (~6–7 hours postexercise). Each participant will be provided with a plastic measuring jug (500 mL) to measure their consumption

at home and to report it back to a researcher over the telephone.

## Postintervention interview

Table 3 lists the domains and assessments that will be repeated at the postintervention interview. Briefly, a home visit will be arranged with each participant after the 6-week intervention to assess his/her general health and physical functioning, including Short Physical Performance Battery, muscle mass (body composition), muscle strength (GS), SF-12, Barthel Index, diet and appetite. Additionally, participants' feedback will be collected at the end of the postintervention interview using a combination of structured multiple-response and standardised open-ended questions. The following themes will be explored: (1) attitudes and barriers to consuming of 2×500 mL milk/control drink intake postexercise (eg, volume of liquid, taste, etc); (2) opinion about milk as a functional food for muscle strength/function; (3) changes in appetite and habitual diet because of milk/control drink intake, and (4) what was liked and disliked about the study (intervention), including motivations and barriers to continue engagement in a local gym. The postintervention interview will be completed within 50 min.

## Statistical methods

As this is a feasibility and acceptability study aimed to inform a larger trial, the focus of data analysis will be descriptive. Using descriptive statistics (percentages, means (SDs)), we will calculate the response rates, the numbers consented and randomised, the retention rate and the number, length and frequency of interviews and RE sessions. Compliance with the milk and control drink intervention will be calculated as a percentage of actual consumption divided by expected consumption over the 6-week intervention. Recording the number of repetitions for each exercise within each RE session and the weight lifted will allow calculation of several indices of training intensity. Mean and SDs (or equivalent) for questionnaire data and assessments will be reported at screening, baseline and postintervention interview. Missing data will be recorded and evaluated.

Participants' experiences and views about the study will be assessed with standardised open-ended questions. These data will be analysed using content analysis.[48] Content analysis is a flexible method for analysing text data. Coding categories will be derived directly from the data and themes will be identified supported with relevant quotations of the participant's perspectives.[48]

The sample size in this pilot study is limited to 30 participants and therefore lacks statistical power for quantitative analysis of the secondary outcomes.

## Dissemination, and patient and public involvement

A consumer panel was organised through the NIHR Research Design Service North East and North Cumbria, UK during the funding application phase of the study in September 2017 to obtain a Patient and Public Involvement (PPI) in the study. We incorporated several panel's suggestions in the study design, including a clear identification of 'community-dwelling' older adults, and the intake of the second dose of milk/control drink within regular diet at home. The following key outputs will contribute to study dissemination and impact. The results of the study will be reported to the funder (NIHR Newcastle Biomedical Research Centre). The funder, sponsor and industry support (Arla) will have no role in the study design, conduct, data analysis, results interpretation or writing. The aim is that at least two peer-reviewed papers will be published in high-impact open-access journals, and the results will be presented at relevant scientific conferences. A lay summary of the main results will be presented to interested participants at a PPI event. A flyer featuring the main results of the study and, if desired, an individual report titled 'My muscle function and strength before and after MIlkMAN' will be prepared for all study participants. Reports with abnormal results (blood pressure, body mass index (BMI), fat mass, Mini Mental State Examination and Geriatric Depression Scale) will be sent to general practices. Regular updates on the study progress will be reported on a publicly accessible website.

## ETHICS AND DATA MONITORING AND MANAGEMENT
### Ethics

The study will be conducted in accordance with the principles of the International Conference for Harmonisation of Good Clinical Practice (European Medicines Agency, 2002). We have amended inclusion criteria for the study, and allowed the inclusion of individuals who have GS or walking speed above the European Working Group on Sarcopenia in Older People (EWGSOP 1) cut-offs.[31]

The study is funded by the National Institute for Health Research Newcastle Biomedical Research Centre, Newcastle University. Arla will provide milk and scientific support related to this nutritional intervention.

### Data monitoring

Throughout the study, the principal investigator (AG) will monitor recruitment, retention and compliance figures with the core research team (AG, CH, LD, TA). The core team will meet regularly to plan and evaluate study's day-to-day activities. Monthly meetings will be organised with the co-investigators (KD, ES, AAS) to update on study management and progress. The core research team and co-investigators will prepare consents, assessments, study protocol and standard operating procedures for: (1) assessments and data reporting; (2) data management; (3) adverse events management and reporting, and (4) staff health risk assessment and safety procedures.

### Adverse events

This is a low-risk study. There is a small chance of transient muscle soreness, gastrointestinal discomfort, metabolic changes and change in appetite. The chief investigator

(TA) is clinically trained to oversee the research process, and the research team is trained in health and safety procedures during data collection. Each participant will be closely monitored and asked about any adverse events occurring at home or in the gym. Any suspected adverse events will be reported to the chief investigator (TA), who will also offer clinical oversight of the study. Any serious adverse events, as evaluated by TA thought to be related to the intervention, will be reported immediately to the study sponsor and relevant ethics committee. Because of the low risk of adverse events, no independent Data Monitoring and Safety Committee will be appointed for this pilot study. The NHS indemnity insurance scheme will apply to cover the potential legal liability cover for harm to participants arising from the research. North Tyneside Council has the public and product liability cover for any potential harm arising from the fitness facility and equipment.

### Data management

Data will be collected and managed in accordance with the EU General Data Protection Regulation (2018). At consent, participants will be assigned a unique study ID that will be used to pseudonymise primary research data collected from interviews and intervention. Identifiable data will be stored separately and will be accessible only to members of the research team who have additional research passport checks approved as part of their research role. Pseudonymised paper-based assessments will be double data entered, and all study data will be stored on secure, fire-wall and password-protected servers of Newcastle University for 5 years.

### Data statement

Technical appendix, statistical code and dataset will be available from the AGE Research Group data manager.

### DISCUSSION
### Strengths and limitations

To our knowledge, this is the first pilot study examining the fidelity of a whole food (milk) combined with RE intervention in community-dwelling older adults living in the UK. The primary aims of the MIlkMAN pilot are to determine the feasibility and acceptability of the intervention in the community, and to provide essential data for planned future substantive research. The secondary aims are exploratory because the pilot lacks power to identify differences in physical functioning between the groups. However, the exploratory findings will be helpful in informing power calculations for the definitive study. The intervention will be conducted under the close supervision of a trained research team including an exercise physiologist and a health psychologist in a local gym with an easy access to older adults. Participants naïve to gym environment will benefit from the familiarisation with RE programme to encourage self-guided continued engagement in the community. A postintervention interview

in the pilot will include the collection of qualitative evidence on the barriers and motivators of community-based interventions. To our knowledge, only one study has investigated the barriers and drivers of compliance with protein-rich diets with RE interventions,[49] and none has included views of older adults about what motivates their willingness and keenness to continue.

This study has several limitations, which will inform the development of the subsequent trial. Physical activity and exercise are consistently reported as positive influences on muscle mass and function in healthy older adults,[16 50] while the evidence for positive effects of protein-rich foods above the effect of RE on muscle in older adults with adequate nutrition and activity levels has been more mixed.[51 52] There may be more benefit for protein supplementation with RE in those with muscle weakness and physical frailty.[20] As the MIlkMAN pilot will enrol 30 participants with relatively healthy muscle, the effect of the intervention is likely to be minimal. To achieve clinically meaningful differences between the groups and to examine the effect of milk above the effect of RE, a larger sample size, longer duration of the intervention and the inclusion of older adults with reduced physical functioning or probable sarcopenia[4] will be necessary. Previous studies have reported difficulties in recruiting older adults with (probable) sarcopenia for various reasons, including the multifaceted nature of muscle health, the variety of muscle-related clinical outcomes relevant to sarcopenia and the lack of routine diagnosis of sarcopenia in clinical practice.[53] However, the universal acceptance of a sarcopenia definition[4 10] and cut-offs for sarcopenia components,[4 10] the availability of sarcopenia screening tools[30] for a rapid assessment of sarcopenia and wider use of GP surgeries (that routinely derive an electronic Frailty Index from data held in healthcare records[54]) for recruitment will increase the potential for enrolling appropriate participants to the larger trial. To reduce the risk of muscle injury, diabetes and exacerbation of any other health risks not covered by the exclusion criteria, this pilot will not recruit older adults with BMI>30 kg/m$^2$. However, in the light of continued debate about the relationship between overweight/obesity and adverse health outcomes,[55] and to maximise the recruitment, the substantive study will consider those with a BMI<35 kg/m$^2$.

We hypothesise that the ratio of protein to fat in whole milk in combination with RE may be beneficial to ageing muscle and superior to skimmed milk for MPS, physical performance and muscle soreness after exercise as observed in younger adults.[27–29] To test this hypothesis and accurately quantify the differences across the groups in the future study, a validated chromatographic analysis of amino acid[56] and fat content[57] in Arla Cravendale milk will be necessary through the scientific support of Arla.

The present study will use BIA to assess body composition in participants preintervention and postintervention. Although BIA has been used widely to estimate lean body mass in community-dwelling older adults via

validated prediction formulas,[58] there are several limitations to the method, including low sensitivity to detect changes in muscle mass and the effect of hydration/dehydration on the analysis.[59] Ultrasound has been proposed as another non-invasive, safe and easy-to-use method suitable for longitudinal monitoring of muscle mass[60] with higher sensitivity compared with BIA. While it requires technical skills,[60] this method may be an appropriate strategy to minimise the limitations associated with BIA to detect changes in muscle mass. In addition, muscle measurements assessed by ultrasound can be compared with anthropometric measures used to estimate RE-induced changes in muscle cross-sectional area, such as thigh circumference and a skinfold thickness,[61] while keeping in mind the limitations of the method in older and obese adults.[60]

To minimise participant burden, the present study will use GS as a measure of overall muscle strength and for minimisation to allow equal distribution of participants with low GS across the groups. However, a future definitive trial will include repeat assessment of 1RM via submaximal testing at baseline and postintervention,[62] to provide a more reliable and internally valid assessment of muscle strength. Repeat assessment of 1RM for all exercises prescribed in the RE programme will enable a more specific evaluation of muscle strength changes following the intervention period.

In summary, this is the first pilot study examining the feasibility and acceptability of whole compared with skimmed milk in combination with RE conducted in a local gym in community-dwelling older adults in the UK. Qualitative data will be collected to inform the future substantive trial, and allow better understanding of the barriers and facilitators of community-based intervention. This pilot study has low statistical power to detect changes in physical functioning between the groups, however, the results will be used to aid the development and refinement of a future clinical trial, including study design, power calculations, recruitment strategy, inclusion and exclusion criteria and outcome measures.

**Author affiliations**
[1]AGE Research Group, Institute of Neuroscience, Newcastle University, Newcastle upon Tyne, UK
[2]Newcastle University Institute for Ageing, Newcastle upon Tyne, UK
[3]NIHR Biomedical Research Centre, Newcastle upon Tyne Hospitals NHS Foundation Trust, Newcastle upon Tyne, UK
[4]Institute of Cellular Medicine, Newcastle University, Newcastle upon Tyne, UK
[5]Human Nutrition Research Centre, Newcastle University, Newcastle upon Tyne, UK

**Acknowledgements** The authors would like to thank the North East and North Cumbria Clinical Research Network.

**Contributors** AG, KD, ES, AAS, TA, LD and CH developed and refined the study protocol. AG, KD, ES, AAS and TA were responsible for study conception and design. AG drafted the manuscript. All coauthors revised the manuscript draft. AG was responsible for the analysis plan. All authors were responsible for critical revision and approved the final version of the manuscript.

**Funding** This project is funded by the National Institute for Health Research (NIHR) Newcastle Biomedical Research Centre (reference number: BH Ref 173606 /

PDB053), Newcastle University and supported by Arla (in-kind (milk) and scientific advice about nutrition).

**Disclaimer** The views expressed are those of the authors and not necessarily those of the NHS or NIHR.

**Competing interests** This study received 'in-kind' contribution from Arla.

**Patient and public involvement statement** A consumer panel organised through the NIHR Research Design Service North East and North Cumbria, UK in September 2017 during the funding application phase.

**Patient consent for publication** Not required.

**Ethics approval** The study approval has been granted by the North East—Newcastle and North Tyneside Research Ethic Committee 1 (REC reference number: 18/NE/0265), and Research and Development (R&D) of the Northumbria Healthcare NHS Foundation Trust (Sponsor).

**Provenance and peer review** Not commissioned; externally peer reviewed.

**ORCID iD**
Antoneta Granic http://orcid.org/0000-0001-9247-899X

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
