## [Reviewer comments · BMJ Open]

ARTICLE DETAILS

TITLE (PROVISIONAL)	Milk and resistance exercise intervention to improve muscle function in community-dwelling older adults at risk of sarcopenia (MilkMAN): protocol for a pilot study
AUTHORS	Granic, Antoneta; Hurst, Christopher; Dismore, Lorelle; Davies, Karen; Stevenson, Emma; Sayer, Avan; Aspray, Terry

VERSION 1 – REVIEW

REVIEWER	Joel T. Cramer University of Nebraska-Lincoln, USA I have current research funding from Abbott Nutrition, the Nebraska Beef Council (NBC), and the National Cattlemen's Beef Association (NCBA) to study metabolic flexibility in older adults, iron/protein intake in youth athletes, and strength as a health indicator in children, respectively. I also am a consultant with Regeneron Pharmaceuticals for research study protocols related to chair stands in the elderly and disabled.
REVIEW RETURNED	30-Apr-2019

GENERAL COMMENTS	Interesting and important study design with aims to examine feasibility and acceptability of 12 sessions of milk and resistance exercise to combat sarcopenia. The investigators are commended for their general attention to detail and well-written protocol. From years of experience designing, analyzing, and publishing clinical trial data on both exercise and nutrition interventions, I would be concerned about the inability to find differences among the groups if the only experimental control is a dietary intervention (milk vs. cold drink). The resistance exercise will be the dominant intervention, which will wash out any dietary interventions, especially in only 6 weeks. Furthermore, the outcome measurements for strength and muscle mass lack internal validity, which will impact external validity. Yet, overall, the study is doable and would be of high value in the field. More detailed thoughts are as follows: 1. Resistance exercise, although minimal over 12 total sessions, will very likely drown out any/all effects of milk. There will be no way to assess the effects of milk alone, without resistance exercise. Therefore, this study is looking for relatively small effects due to the milk vs. cold drink interventions, since all subjects participate in the resistance exercise intervention. Relatively small samples sizes in each group (n=10) will be unlikely to show meaningful, clinical differences among or between groups. However, this may not be problematic, since 6 weeks of any dietary intervention will be minimal at best.
---

2. A recent systematic review indicated that exercise is the primary driver of muscle mass and function in healthy subjects aged 60 years and older. Dietary interventions have limited meaningful effects, especially over 6 weeks. The investigators are encouraged to consider this aspect in conjunction with the participants being "healthy older adults." When participants are relatively well-fed and relatively active (i.e., healthy), there are no real deficits to overcome. However, if the subjects are indeed already sarcopenic (i.e., low handgrip strength, low gait speed, Skeletal Mass Index cutoffs, etc.) and/or malnourished (i.e., Subjective Global Assessment, etc.), there are inherent deficits to overcome, which will improve the odds of study intervention success. Based on the inclusion/exclusion criteria of the present study, such deficits are not necessarily present, which may limit the success of the interventions, and likely lower the probability of seeing an interaction effect. See Beaudart et al. (2017) Nutrition and physical activity in the prevention and treatment of sarcopenia: systematic review. *Osteoporos Int*, Jun, 28(6):1817-1833.

3. Since the primary/secondary hypotheses deal with milk protein and milk fat differences among group interventions, is there a way to assess the amino acid and fatty acid contents of the milk interventions throughout the 6 weeks? After all, this is the totality of differences among the 3 groups in this study; thus, it would be appropriate to accurately quantify those differences.

4. What is the rationale for incorporating an exclusion criteria of BMI ≥ 30 ? This may limit recruitment a bit. Furthermore, there are continued debates regarding the risk of mortality and obesity. See Flegal et al. (2019) Flawed methods and inappropriate conclusions for health policy on overweight and obesity: the Global BMI Mortality Collaboration meta-analysis. *Journal of Cachexia, Sarcopenia and Muscle* 10:9-13. Would the authors consider increasing the BMI exclusion to > 35 ?

5. Table 1 on inclusion/exclusion criteria are not entirely consistent with the text on page 9, lines 51-57, and page 10, lines 3-6. If grip strength and gait speed will not be used as inclusion/exclusion criteria, but will be used for categorization or delimiting the sample, the rationale and criteria for numbers of participants with "high" grip strength versus "low" grip strength should also be described.

6. The citation used for the grip strength and gait speed criteria for high versus low should be updated. The grip strength cutoffs have changed. See: Cruz-Jentoft AJ, Bahat G, Bauer J, Boirie Y, Bruyère O, Cederholm T, Cooper C, Landi F, Rolland Y, Sayer AA, Schneider SM, Sieber CC, Topinkova E, Vandewoude M, Visser M, Zamboni M; Writing Group for the European Working Group on Sarcopenia in Older People 2 (EWGSOP2), and the Extended Group for EWGSOP2. Sarcopenia: revised European consensus on definition and diagnosis. *Age Ageing*. 2019 Jan 1;48(1):16-31. doi: 10.1093/ageing/afy169. PubMed PMID: 30312372; PubMed Central PMCID: PMC6322506.

7. The investigators should be commended for their resistance exercise protocol details. Very nicely done. Page 14, line 12: Is 45-60 minutes of exercise duration accurate for 2-4 sets of only four separate exercises? In most of our previous studies, this would take approximately 20-30 minutes to complete, even in a small group setting.

	8. What are the criteria for successfully completing the resistance exercise program (i.e., how many exercises must be completed for each participant to be considered compliant)? 9. Measuring muscle mass differences among groups with BIA will be difficult. BIA is often criticized as an invalid measure of body composition, but a valid measure of total body water. The sensitivity of BIA measurements is also limited. Is there any way the investigators could consider using a B-mode ultrasound for muscle thickness assessments? Ultrasound is arguably quicker and simpler than BIA, and has much greater validity and sensitivity for finding small differences in muscle mass between the milk and control groups. Chumlea WC, Guo SS, Kuczmarski RJ, Vellas B. Bioelectric and anthropometric assessments and reference data in the elderly. J Nutr. 1993 Feb;123(2 Suppl):449-53. doi: 10.1093/jn/123.suppl_2.449. Review. PubMed PMID: 8429402. Thompson DL, Thompson WR, Prestridge TJ, Bailey JG, Bean MH, Brown SP, McDaniel JB. Effects of hydration and dehydration on body composition analysis: a comparative study of bioelectric impedance analysis and hydrodensitometry. J Sports Med Phys Fitness. 1991 Dec;31(4):565-70. PubMed PMID: 1806735. Pearman P, Hunter G, Hendricks C, O'Sullivan P. Comparison of hydrostatic weighing and bioelectric impedance measurements in determining body composition pre- and postdehydration. J Orthop Sports Phys Ther. 1989;10(11):451-5. PubMed PMID: 18796946. 10. Measurements of grip strength as the "strength" outcome are not specific to this study's intervention. Although grip strength is convenient and functional, there are no exercises in this protocol designed specifically to increase grip strength. Since there are 4 resistance exercises (chest press, seated row, leg extension, and leg curls), the investigators are encouraged to select measurements of strength that will more internally valid and more likely be influenced by these exercises. For example, the 1RM estimated by a submaximal 5RM test would allow investigators to see strength increases in all 4 exercises on the same equipment used for the exercises themselves. 11. A similar argument could be made for calf circumference. There are no exercises to influence the calf muscles per se. Would the investigators consider thigh circumference as a more internally valid measurement instead? Also consider including a skinfold at the same thigh placement as the circumference measurement (50% distance between ASIS and patella) on the anterior thigh. Using thigh circumference and a single skinfold at that location allows an estimate of muscle cross-sectional area. See: DeFreitas et al. (2010) A comparison of techniques for estimating training-induced changes in muscle cross-sectional area. J Strength Cond Res, Sep; 24(9):2383-2389.
--	--

REVIEWER	Lars Holm University of Birmingham
REVIEW RETURNED	24-Jun-2019

GENERAL COMMENTS	To the authors
----------------

	I think that this study appears well-considered a well-designed. I have some comments to be considered: Look at the work recently done by Hamarsland H and Raastad (PMID 30157103) and Aas SN and Raastad (PMID 31183750) from Oslo on milk protein intake in young and older people, respectively. Why were 6 weeks intervention in this feasibility study chosen? Because a long-term study will be of the same length? I think that adherence is very much dependent on the length of the intervention period and hence, should be equal between feasibility studies and the 'real' study! The length of the big study should of course be set dependent on the primary outcome. If it is muscle mass development or something similar, I would think that 6 weeks is too short an intervention period. Minor comments: P.5, line 24: Prevalence of sarcopenia depends on the definition used. P. 14, line 39-56: How often will the resistance training intensity be evaluated. Be aware that it is hard to get older unaccustomed people to train hard!!
--	---

VERSION 1 – AUTHOR RESPONSE

Reviewer 1.

Interesting and important study design with aims to examine feasibility and acceptability of 12 sessions of milk and resistance exercise to combat sarcopenia. The investigators are commended for their general attention to detail and well-written protocol.

From years of experience designing, analyzing, and publishing clinical trial data on both exercise and nutrition interventions, I would be concerned about the inability to find differences among the groups if the only experimental control is a dietary intervention (milk vs. cold drink). The resistance exercise will be the dominant intervention, which will wash out any dietary interventions, especially in only 6 weeks. Furthermore, the outcome measurements for strength and muscle mass lack internal validity, which will impact external validity. Yet, overall, the study is doable and would be of high value in the field. More detailed thoughts are as follows:

Author's response:

We would like to thank the reviewer for the encouraging words and the time commitment reviewing our manuscript. We have carefully reviewed and addressed the reviewer's comments below and in the manuscript.

1. Resistance exercise, although minimal over 12 total sessions, will very likely drown out any/all effects of milk. There will be no way to assess the effects of milk alone, without resistance exercise. Therefore, this study is looking for relatively small effects due to the milk vs. cold drink interventions, since all subjects participate in the resistance exercise intervention. Relatively small samples sizes in each group (n=10) will be unlikely to show meaningful, clinical differences among or between groups. However, this may not be problematic, since 6 weeks of any dietary intervention will be minimal at best.

Author's response/action:

We agree with the reviewer that the effect of dietary intervention (whole and skimmed milk) in this pilot will be minimal and so may be undetectable because of the size of the study arms and the fact that all participants will be allocated to the resistance exercise (RE) intervention.

The results of the pilot will aid the development and refinement of the protocol for the subsequent trial that will assess the effect of milk + RE on physical performance in older adults. The focus of the pilot is the fulfilment of the primary aim: the feasibility and acceptability of the intervention in the community (a local gym), including any difficulties with the intake of 2 x 500ml of liquid twice a week over 6 consecutive weeks. Because the sample size is limited to 30 participants (10 in each group) and the rationale for it aided by the primary aims, we agree with the reviewer that we do not have statistical power for a proper quantitative analysis

of the secondary outcomes. Therefore, we will explore the secondary aims to inform the design (including the sample size, study duration, outcomes, etc.) of the future trial.

2. A recent systematic review indicated that exercise is the primary driver of muscle mass and function in healthy subjects aged 60 years and older. Dietary interventions have limited meaningful effects, especially over 6 weeks. The investigators are encouraged to consider this aspect in conjunction with the participants being "healthy older adults." When participants are relatively well-fed and relatively active (i.e., healthy), there are no real deficits to overcome. However, if the subjects are indeed already sarcopenic (i.e., low handgrip strength, low gait speed, Skeletal Mass Index cutoffs, etc.) and/or malnourished (i.e., Subjective Global Assessment, etc.), there are inherent deficits to overcome, which will improve the odds of study intervention success. Based on the inclusion/exclusion criteria of the present study, such deficits are not necessarily present, which may limit the success of the interventions, and likely lower the probability of seeing an interaction effect. See Beaudart et al. (2017) Nutrition and physical activity in the prevention and treatment of sarcopenia: systematic review. Osteoporos Int, Jun, 28(6):1817-1833.

Author's response/action:

We agree with the reviewer that the effect of study intervention would be greater in older adults who have inherent deficits to overcome such as sarcopenia (elements of sarcopenia) or malnutrition. However, we do not aim to treat sarcopenia but to recruit older adults who did not have any current deficits (ageing 'healthy'), but who may experience muscle health-related difficulties in the future (i.e. may have short or medium term risk of sarcopenia). We believe that to foster healthy muscle ageing and minimise the risk of sarcopenia in older adults, interventions should start prior to muscle mass and function falling below the proposed cutoffs for its diagnosis.

To justify the inclusion of 'healthy' older adults we have added the following rationale for this population in the Introduction:

'This emphasises the need for sustainable preventive measures aimed to preserve and optimise muscle health and function in a rapidly ageing population *before the onset of difficulties leading to or exacerbating the risk of sarcopenia*' (page 5, paragraph 2 in the manuscript with 'tracked changes');

and in the Study population and recruitment section:

'Therefore, the study will recruit older adults with some deficits in muscle health and those without for whom it is determined to be safe to participate in the study (primary aims), and hypothesised to benefit from the intervention regardless of these deficits (secondary aims)' (page 10 in the manuscript with 'tracked changes').

Also, to facilitate the completion of the primary aims, we have anticipated that, for safety reasons, the inclusion of frailer older adults would pose additional challenges.

3. Since the primary/secondary hypotheses deal with milk protein and milk fat differences among group interventions, is there a way to assess the amino acid and fatty acid contents of the milk interventions throughout the 6 weeks? After all, this is the totality of differences among the 3 groups in this study; thus, it would be appropriate to accurately quantify those differences.

Author's response/action:

We thank the reviewer for this suggestion. The reviewer is correct in stating that for our secondary hypothesis the totality of differences across the three interventions deals with the amino acid and fatty acid content in milk. Such data for this pilot protocol is not available, but we welcome the suggestion for the future publication. For the future trial, Arla has expressed an interest in providing scientific support, which could include the analysis of the amino acid / fatty acid average content of Arla Cravendale milk from British farms, in addition to the support they offered to the MIikMAN pilot.

4. What is the rationale for incorporating an exclusion criteria of BMI ≥ 30 ? This may limit recruitment a bit. Furthermore, there are continued debates regarding the risk of mortality and obesity. See Flegal et al. (2019) Flawed methods and inappropriate conclusions for health policy on overweight and obesity: the Global BMI Mortality Collaboration meta-analysis. Journal of Cachexia, Sarcopenia and Muscle 10:9-13. Would the authors consider increasing the BMI exclusion to > 35 ?

Authors' response/action:

We thank the reviewer for pointing out that there are limitations to WHO cutoffs for obesity in relation to health outcomes in older adults. The rationale for BMI ≥ 30 is our concern about participants' health and safety, including potential difficulties in using resistance exercise machines and health concerns relating to diet (fat content in whole milk, carbohydrate content in supplemented control drink). These were considered to be important safeguards to limit the risk of muscle injury, diabetes, and exacerbation of any other risks not covered by the inclusion/exclusion criteria. However, we may consider increasing the BMI exclusion to >35 in the future.

5. Table 1 on inclusion/exclusion criteria are not entirely consistent with the text on page 9, lines 51-57, and page 10, lines 3-6. If grip strength and gait speed will not be used as inclusion/exclusion criteria, but will be used for categorization or delimiting the sample, the rationale and criteria for numbers of participants with "high" grip strength versus "low" grip strength should also be described.

Authors' response/action:

We thank the reviewer for pointing out the inconsistency in inclusion/exclusion criteria in the text and Table 1, which we have corrected. We reworded the third paragraph in the Study population and recruitment section (page 9) to clarify that in addition to exclusion and inclusion criteria, we will assess participants muscle strength (grip strength, GS) and walking speed at the screening interview. GS will be used for minimisation as described in the manuscript. Both those with 'low' and 'high' GS will be included in the study without determining the numbers in each category a priori (page 10 in the manuscript with 'tracked changes'). The evaluation of the walking speed at participants' home serves to confirm the criteria described in the Table 1 as 'an individual who the research team (exercise physiologist) evaluates as not suitable for the intervention because of safety reasons'.

6. The citation used for the grip strength and gait speed criteria for high versus low should be updated. The grip strength cutoffs have changed. See: Cruz-Jentoft AJ, Bahat G, Bauer J, Boirie Y, Bruyère O, Cederholm T, Cooper C, Landi F, Rolland Y, Sayer AA, Schneider SM, Sieber CC, Topinkova E, Vandewoude M, Visser M, Zamboni M; Writing Group for the European Working Group on Sarcopenia in Older People 2 (EWGSOP2), and the Extended Group for EWGSOP2. Sarcopenia: revised European consensus on definition and diagnosis. *Age Ageing*. 2019 Jan 1;48(1):16-31. doi: 10.1093/ageing/afy169. PubMed PMID: 30312372; PubMed Central PMCID: PMC6322506.

Author's response/action:

We thank the reviewer for this comment. We have included the updated version of the EWGSOP definition for sarcopenia in our original manuscript (reference 4), and added a recently published seminar paper from the authors to the reference list (now reference 10). However, for the minimisation to the study groups and balancing of the recruitment, we have used the old criteria to be consistent with the previous studies conducted by our group that helped in designing the pilot. In addition, during the application phase for the study the new cutoffs were not available. However, we will use the EWGSOP2 algorithm for the analysis of the pilot study data and for future studies.

7. The investigators should be commended for their resistance exercise protocol details. Very nicely done. Page 14, line 12: Is 45-60 minutes of exercise duration accurate for 2-4 sets of only four separate exercises? In most of our previous studies, this would take approximately 20-30 minutes to complete, even in a small group setting.

Author's response/action:

Thank you for this comment and for recognising the strength in our protocol. We allowed for a longer duration of exercise slots because of the intervention setting in a local gym. Each participant will receive a free gym membership and will exercise during off-peak hours. However, the availability of the resistance exercise machines cannot be guaranteed, and we have to leave a very conservative time for exercise time slots. For those who are unexperienced, we have allowed for additional time if demonstration of the exercise programme needs to be repeated. The 45-60 minutes time slot also includes warm-up and cool-down exercises, and gym orientation (first session).

However, we recognise that the described exercises can be completed within 30 minutes, and we added a sentence to indicate this possibility (page 15, first paragraph in the manuscript with 'tracked changes').

8. What are the criteria for successfully completing the resistance exercise program (i.e., how many exercises must be completed for each participant to be considered compliant)?

Author's response/action:

We have added the following criteria: completing at least 10 sessions out of 12 (page 15, paragraph 2 in the manuscript with 'tracked changes').

9. Measuring muscle mass differences among groups with BIA will be difficult. BIA is often criticized as an invalid measure of body composition, but a valid measure of total body water. The sensitivity of BIA measurements is also limited. Is there any way the investigators could consider using a B-mode ultrasound for muscle thickness assessments? Ultrasound is arguably quicker and simpler than BIA, and has much greater validity and sensitivity for finding small differences in muscle mass between the milk and control groups.

Chumlea WC, Guo SS, Kuczmarski RJ, Vellas B. Bioelectric and anthropometric assessments and reference data in the elderly. J Nutr. 1993 Feb;123(2 Suppl):449-53. doi: 10.1093/jn/123.suppl_2.449. Review. PubMed PMID: 8429402.

Thompson DL, Thompson WR, Prestridge TJ, Bailey JG, Bean MH, Brown SP, McDaniel JB. Effects of hydration and dehydration on body composition analysis: a comparative study of bioelectric impedance analysis and hydrodensitometry. J Sports Med Phys Fitness. 1991 Dec;31(4):565-70. PubMed PMID: 1806735.

Pearman P, Hunter G, Hendricks C, O'Sullivan P. Comparison of hydrostatic weighing and bioelectric impedance measurements in determining body composition pre- and postdehydration. J Orthop Sports Phys Ther. 1989;10(11):451-5. PubMed PMID: 18796946.

Author's response/action:

We acknowledge the reviewer's comment about the limitations of BIA despite the formulas^{1,2} being developed in older adults to improve its sensitivity compared to DXA.

1. Kyle UG, Genton L, Hans D, Pichard C. Validation of a bioelectrical impedance analysis equation to predict appendicular skeletal muscle mass (ASMM). Clin Nutr (Edinburgh, Scotland) 2003;22(6):537e43.
2. Vermeiren S, Beckwée D, Vella-Azzopardi R, et al. Evaluation of appendicular lean mass using bio impedance in persons aged 80+: A new equation based on the BUTTERFLY-study. Clin Nutr. 2019;38(4):1756-1764.

We agree with the reviewer that a B-mode ultrasound for muscle thickness assessments may be a better alternative that is gaining popularity in research. We welcome this idea for the larger trial.

For the pilot, we used the experiences from our research group (e.g. Newcastle 85+ Study¹, MASS_pilot², and LACE³) in conducting body composition analysis for participants in the community to estimate lean body mass via prediction formulas, which is quick and cost-effective. This pilot has limited costs allocated to staff and equipment, and we assume that the B-mode ultrasound will need an experienced ultrasonographer.

1. Dodds RM, Granic A, Davies K, Kirkwood TB, Jagger C, Sayer AA. Prevalence and incidence of sarcopenia in the very old: findings from the Newcastle 85+ Study. J Cachexia Sarcopenia Muscle. 2017;8(2):229-237.
2. Dodds RM, Davies K, Granic A, Hollingsworth KG, Warren C, Gorman G, Turnbull DM, Sayer AA. Mitochondrial respiratory chain function and content are preserved in the skeletal muscle of active very old men and women. Exp Gerontol. 2018;113:80-85.
3. Band MM, Sumukadas D, Struthers AD, Avenell A, Donnan PT, Kemp PR, Smith KT, Hume CL, Hapca A, Witham MD. Leucine and ACE inhibitors as therapies for sarcopenia (LACE trial): study protocol for a randomised controlled trial. Trials. 2018;19(1):6.

We acknowledge that this pilot has limited ability to detect the differences in muscle mass (secondary aim). Please also see our comment under the point 1.

10. Measurements of grip strength as the "strength" outcome are not specific to this study's intervention. Although grip strength is convenient and functional, there are no exercises in this protocol designed specifically to increase grip strength. Since there are 4 resistance exercises (chest press, seated row, leg extension, and leg curls), the investigators are encouraged to select measurements of strength that will more internally valid and more likely be influenced by these exercises. For example, the 1RM estimated by a submaximal 5RM test would allow investigators to see strength increases in all 4 exercises on the same equipment used for the exercises themselves.

Author's response/action:

We thank the reviewer for this comment, which we will consider for the larger study. In the pilot, we will use GS for two reasons: (a) to identify those potentially at risk of sarcopenia (as a part of sarcopenia algorithm) for minimisation across the intervention groups, and (b) as an outcome of global muscle function¹.

We will not evaluate individual muscle function to avoid extra burden to participants. However, for each participant we will record weight lifted per session in a training log containing diagrams and short instructions about the exercises completed. This log will encourage the participants to continue the exercises on their own.

The secondary aim of the study will explore changes in muscle function (GS as a part of it) across the groups to inform the future trial, although we recognise that the pilot lacks statistical power for a proper quantitative analysis.

1. Bohannon RW. Muscle strength: clinical and prognostic value of hand-grip dynamometry. *Curr Opin Clin Nutr Metab Care*. 2015;18(5):465-70.

11. A similar argument could be made for calf circumference. There are no exercises to influence the calf muscles per se. Would the investigators consider thigh circumference as a more internally valid measurement instead? Also consider including a skinfold at the same thigh placement as the circumference measurement (50% distance between ASIS and patella) on the anterior thigh. Using thigh circumference and a single skinfold at that location allows an estimate of muscle cross-sectional area. See: DeFreitas et al. (2010) A comparison of techniques for estimating training-induced changes in muscle cross-sectional area. *J Strength Cond Res*, Sep; 24(9):2383-2389.

Author's response/action:

We thank the reviewer for these suggestions, and in the future study we will consider thigh circumference as a measure more specific to the study's intervention. In the pilot, we will use calf circumference only at baseline and primarily as a measure associated with malnutrition (i.e. calf circumference is a part of Mini Malnutrition Assessment)¹⁻³.

1. Cereda E. Mini nutritional assessment. *Curr Opin Clin Nutr Metab Care*. 2012;15(1):29-41.

2. Maeda K, Koga T, Nasu T, Takaki M, Akagi J. Predictive accuracy of calf circumference measurements to detect decreased skeletal muscle mass and European Society for Clinical Nutrition and Metabolism-Defined malnutrition in hospitalized older patients. *Ann Nutr Metab*. 2017;71(1-2):10-15.

3. Tsai AC, Chang TL, Wang YC, Liao CY. Population-specific short-form mini nutritional assessment with body mass index or calf circumference can predict risk of malnutrition in community-living or institutionalized elderly people in Taiwan. *J Am Diet Assoc*. 2010;110(9):1328-34.

We would like to thank the reviewer for the detailed attention to the manuscript.

Reviewer 2.

1. I think that this study appears well-considered a well-designed.

Author's response/action:

We thank the reviewer for recognising some strength of our study.

I have some comments to be considered:

2. Look at the work recently done by Hamarsland H and Raastad (PMID 30157103) and Aas SN and Raastad (PMID 31183750) from Oslo on milk protein intake in young and older people, respectively.

Why were 6 weeks intervention in this feasibility study chosen? Because a long-term study will be of the same length? I think that adherence is very much dependent on the length of the intervention period and hence, should be equal between feasibility studies and the ‘real’ study! The length of the big study should of cause be set dependent on the primary outcome. If it is muscle mass development or something similar, I would think that 6 weeks is too short an intervention period.

Author’s response/action:

We would like to thank the reviewer for pointing out the two recent studies involving strength training with milk in younger and older adults, which we read with great interest.

We have opted for the 6-week intervention for the following reasons. As stated in our response to Reviewer 1, a primary aim of the pilot is to gather information for the development and refinement of the protocol for the subsequent trial. The duration of the trial will depend on the pilot findings, especially the acceptability and feasibility of the intervention in the community. Specifically, one important concern is the ability to continue consuming 1 litre of milk or control drink a day twice a week over the period of 6 weeks, including compliance, tolerance, appetite suppression, and adverse health events resulting from the consumption. Next, we will incorporate the results of qualitative research (post-intervention interview), which will probe for motivations and barriers to continue engagement in the intervention (nutrition + exercise) after the study, and the post hoc changes in participants’ behaviour (continuing engagement). Although we will only explore and describe the differences across the intervention groups in physical functioning, these findings will aid the design of the future trial. Taken together, the results will help us to determine whether an intervention longer than 6 weeks or blocks of 6 weeks will be required for the future trial.

3. Minor comments:

P.5, line 24: Prevalence of sarcopenia depends on the definition used.

Author’s response/action:

We added the following text and a reference (reference 10) to indicate that the prevalence of sarcopenia in a population is dependent on sarcopenia definition.

‘The prevalence of sarcopenia increases with advancing age—and *although dependent on the algorithm used to define sarcopenia*¹⁰ (page 5 in the manuscript with ‘tracked changes’).

Cruz-Jentoft AJ, Sayer AA. Sarcopenia. Lancet 2019;393: 2636-46.

4. P. 14, line 39-56: How often will the resistance training intensity be evaluated. Be aware that it is hard to get older unaccustomed people to train hard!!

Author’s response/action:

Thank you for your comment. Resistance training intensity will be monitored and recorded following each exercise session using measures of external training load (e.g. repetitions completed, volume load) and internal training load (ratings of perceived exertion)¹. This information will be used to ensure appropriate progressive overload throughout the exercise intervention. Further information has been added to the manuscript to clarify this (page 15; paragraph 2 in the manuscript with ‘tracked changes’). The short duration of the training programme (6 weeks) means that we will not perform any repeat assessment of 1RM during the intervention period, but the measurements will be considered for the future trial of longer duration.

1. Hurst C, Weston KL, Weston M. The effect of 12 weeks of combined upper-and lower-body high-intensity interval training on muscular and cardiorespiratory fitness in older adults. Aging Clin Exp Res 2019;31:661–71.

We thank the reviewers for the encouraging words and the time commitment reviewing our manuscript.

VERSION 2 – REVIEW

REVIEWER	Joel T. Cramer University of Nebraska-Lincoln, United States
REVIEW RETURNED	26-Jul-2019

GENERAL COMMENTS	I have read through the authors' responses. The paraphrased response to nearly all 11 original questions was "We thank the reviewer, but no we will not make the change. Instead, we will consider it for future studies." Unfortunately, the inability/unwillingness to adapt the protocol to make it better leaves me with little confidence in the ability of this study to answer the research questions or contribute to our knowledge of sarcopenia. In most cases, the authors agreed with my comments/questions, but did little (if anything) to revise the protocol. I would encourage proposals of studies that advance our knowledge of sarcopenia and are carefully designed to answer research questions with valid, reliable, and sensitive measurements. I commend the authors on their writing style, and their commitment to this particular design, but there are flaws in the design that the investigators could overcome, but chose not to.
--

REVIEWER	Lars Holm University of Birmingham
REVIEW RETURNED	05-Aug-2019

GENERAL COMMENTS	The authors have replied my comments and made changes in the manuscript accordingly. I have no further comments or questions.
---

VERSION 2 – AUTHOR RESPONSE

Reviewer 1.

I have read through the authors' responses. The paraphrased response to nearly all 11 original questions was "We thank the reviewer, but no we will not make the change. Instead, we will consider it for future studies." Unfortunately, the inability/unwillingness to adapt the protocol to make it better leaves me with little confidence in the ability of this study to answer the research questions or contribute to our knowledge of sarcopenia. In most cases, the authors agreed with my comments/questions, but did little (if anything) to revise the protocol. I would encourage proposals of studies that advance our knowledge of sarcopenia and are carefully designed to answer research questions with valid, reliable, and sensitive measurements. I commend the authors on their writing style, and their commitment to this particular design, but there are flaws in the design that the investigators could overcome, but chose not to.

Author's response/action:

We would like to thank the Reviewer for the constructive criticism of the pilot study protocol, and we recognise the importance of the concerns raised about the study weaknesses. We have tried to address all 11 points raised by the Reviewer as thoroughly as possible and it has certainly not been our intent to take them lightly or to ignore them. However, we also appreciate that our responses fell short of the expected changes in the study protocol. We submitted the manuscript to the journal recognising, as it states in the guidance to authors, that "Protocol manuscripts should report planned or ongoing research studies". At the time of the manuscript submission, the pilot was already in the first stage of data collection (i.e. 10 participants were minimised to the study arms and started the intervention). Furthermore, strict financial and time constraints would prevent any substantial changes to the study design, which would also require substantial amendments to be submitted to the research ethics committee.

The focus of this pilot is to determine the feasibility and acceptability of the intervention in the community (a local leisure facility), including any difficulties with the intake of 1 l of milk/control drink twice a week

over 6 consecutive weeks. We will use the findings from this pilot together with the helpful comments from the Reviewer in due course to design the main MilkMAN study and overcome the existing design flaws. We are confident that this will lead to a trial that will advance the knowledge of sarcopenia.

We have outlined the strengths and weaknesses of the pilot study in the Discussion (pages 20-23), incorporated the Reviewer's suggestions, and included five references provided previously by the Reviewer. We included the following points and discussed how they will be addressed in the future trial:

1. *Weaknesses of the study design:* (a) short duration; (b) small sample size; (c) inclusion and exclusion criteria (i.e. lack of individuals who are pre-sarcopenic or sarcopenic; exclusion of obese older adults [BMI \geq 30]), and (d) small effect of dietary intervention above the effect of exercise.

2. *Outcomes and measurements:* (a) weaknesses of BIA for measurement of change in muscle mass; (b) new sarcopenia algorithms and grip strength cut-offs for pre-sarcopenia; (c) measurements of strength changes using test internal to resistance exercise programme (e.g. prediction of the 1-RM strength from a 4-5 RM submaximal strength test); (d) thigh circumference and a skinfold as measures of muscle cross-sectional area, and (e) analysis of the amino acid and fat content in whole and skimmed milk.

We thank the Reviewer for their thorough attention to our manuscript.

Reviewer 2.

The authors have replied my comments and made changes in the manuscript accordingly. I have no further comments or questions.

Author's response/action:

We thank the Reviewer for their attention to our manuscript.